# CAN MLLMS UNDERSTAND THE DEEP IMPLICATION BEHIND CHINESE IMAGES?

## ABSTRACT

As the capabilities of Multimodal Large Language Models (MLLMs) continue to improve, the need for higher-order capability evaluation of MLLMs is increasing. However, there is a lack of work evaluating MLLM for higher-order perception and understanding of Chinese visual content. To fill the gap, we introduce the **C**hinese **I**mage **I**mplication understanding **Bench**mark, **CII-Bench**, which aims to assess the higher-order perception and understanding capabilities of MLLMs for Chinese images. CII-Bench stands out in several ways compared to existing benchmarks. Firstly, to ensure the authenticity of the Chinese context, images in CII-Bench are sourced from the Chinese Internet and manually reviewed, with corresponding answers also manually crafted. Additionally, CII-Bench incorporates images that represent Chinese traditional culture, such as famous Chinese traditional paintings, which can deeply reflect the model's understanding of Chinese traditional culture. Through extensive experiments on CII-Bench across multiple MLLMs, we have made significant findings. Initially, a substantial gap is observed between the performance of MLLMs and humans on CII-Bench. The highest accuracy of MLLMs attains 64.4%, where as human accuracy averages 78.2%, peaking at an impressive 81.0%. Subsequently, MLLMs perform worse on Chinese traditional culture images, suggesting limitations in their ability to understand high-level semantics and lack a deep knowledge base of Chinese traditional culture. Finally, it is observed that most models exhibit enhanced accuracy when image emotion hints are incorporated into the prompts. We believe that CII-Bench will enable MLLMs to gain a better understanding of Chinese semantics and Chinese-specific images, advancing the journey towards expert artificial general intelligence (AGI).

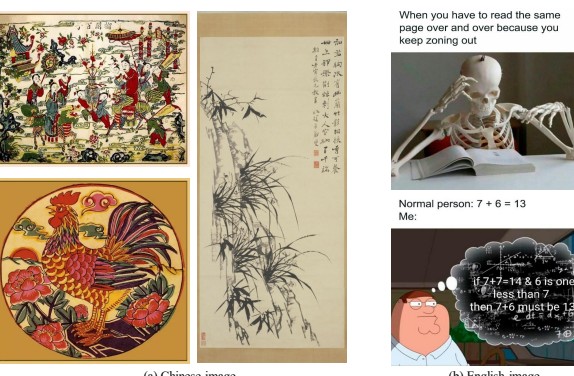

(a) Chinese image      (b) English image

Figure 1: Comparision of Chinese and English image implications. Chinese images often embody richer scenes and deeper implications with Chinese traditional culture compared with the straight-forward and explicit symbolism in English images.

## 1 INTRODUCTION

With the rapid advancement of artificial intelligence, Multimodal Large Language Models (MLLMs) (Liu et al., 2023b; Li et al., 2023c; Ye et al., 2023; Tong et al., 2024) have demonstrated exceptional

performance across various domains, including natural language processing (Chowdhary & Chowdhary, 2020; Luo et al., 2024; Zhang et al., 2024a) and computer vision (Lu et al., 2022; Li et al., 2023b;a; Xu et al., 2023; Fu et al., 2023; Cai et al., 2023; Zhang et al., 2023; Chen et al., 2024b; Jin et al., 2024). These models are not only capable of processing and generating text but also excel at integrating and interpreting information across multiple modalities, such as images, sound, and video. However, despite the significant progress made in tasks like image recognition and generation, a crucial research question remains: Can these models truly understand and interpret images that have deep implications? (Liu et al., 2024b) construct an English image implication understanding dataset, II-Bench, and the experiments on MLLMs and human subjects reveal a substantial gap in the models' higher-order perception abilities, particularly in nuanced emotional understanding and profound meaning extraction, when compared to humans. Unfortunately, the rapid advancement of MLLMs has led to significant performance improvements. For instance, Claude-3.5-Sonnet has achieved an impressive accuracy of 80.9% on II-Bench, approaching the average human accuracy of 90.3%. This progress underscores the need for more challenging benchmarks that incorporate richer scenes and deeper implications to continue pushing the boundary of image implication understanding task.

In contrast to English images, Chinese images often embody richer scenes (Xu, 2023) and deeper implications as Figure 1 shows. For instance, Chinese traditional landscape paintings not only depict natural scenery but also convey profound philosophical concepts, such as the harmony between humans and nature, through artistic techniques like the interplay of void and solid, the use of negative space, and the brushwork. As the famous Chinese poet Su Shi noted, "Poetry and painting share the same essence, embodying both craftsmanship and purity". The depth of Chinese images lies not only in their aesthetic appeal but also in the underlying spirit and philosophy they express. Similarly, New Year paintings, as a significant carrier of Chinese traditional culture, typically use symbolism and implication to convey wishes for good fortune, prosperity, and peace. Unlike the directness often found in English imagery, Chinese images emphasize the creation of atmosphere and subtle expression, requiring viewers to possess certain cultural knowledge to accurately grasp their meanings. This cultural disparity leads to significant differences in the modes of expression and meaning conveyed between Chinese and English images, highlighting the need to consider cultural context when evaluating the capability of MLLMs to understand the deep implications of images.

To address this gap, we develop CII-Bench, a benchmark designed to comprehensively test the higher-order perception, reasoning, and understanding abilities of models within a Chinese context. This benchmark allows us to gain a clearer understanding of these models' interpretive capacities, offering new insights into their application in cross-cultural environments, and thus advancing the research and development of MLLMs.

As illustrated in Figure 2, CII-Bench comprises 698 images and 800 multiple-choice questions spanning six domains: Life, Art, Society, Politics, Environment, and Chinese Traditional Culture. Moreover, to ensure diversity, CII-Bench includes six types of images: Illustration, Meme, Poster, Single-panel Comic, Multi-panel Comic, and Painting. By employing images of various types and from different domains, the benchmark provides a more robust evaluation of models' comprehension and reasoning abilities.

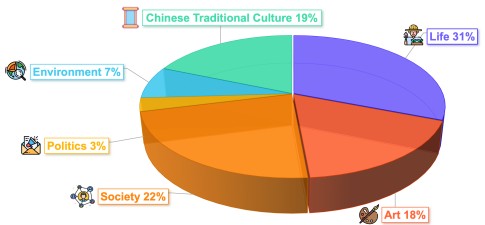

Figure 2: Composition of CII-Bench.

We conduct extensive experiments to evaluate CII-Bench on MLLMs that support Chinese and deeply evaluate the model's grasp of Chinese traditional culture. Our key contributions are as follows:

- We introduce CII-Bench, the first benchmark designed to assess the understanding of meanings in Chinese images, which poses a significant challenge to current MLLMs.
- We design a comprehensive evaluation metric based on GPT-4o to evaluate Chinese traditional culture. This metric aligns more closely with human annotations and is better suited for evaluating Chinese traditional painting.
- Our experimental findings are as follows: (1) There is a notable performance gap between MLLMs and humans. Models demonstrate the highest accuracy of 64.4%, while human accuracy average at 78.2% and best at 81.0%. (2) Closed-source models generally outperform

open-source models, but the best-performing open-source model surpasses the top closed-source model, with a difference of more than 3%. (3) Models perform significantly worse in Chinese traditional culture compared to other domains, indicating that current models still lack sufficient understanding of Chinese culture. Further analysis shows that GPT-4o can only observe the surface-level information, it's difficult to deeply interpret the complex cultural elements contained in Chinese traditional painting. (4) Incorporating image emotion hints into prompts generally improves model scores, indicating that models struggle with emotional understanding, leading to misinterpretation of the implicit meanings in the images.

## 2 RELATED WORK

### 2.1 MULTIMODAL LARGE LANGUAGE MODELS

With the rapid development of large language models (LLMs) (Aakanksha et al., 2022; Won et al., 2022; Chiang et al., 2023; Touvron et al., 2023; OpenAI, 2023a;b; Team, 2024; Cai et al., 2024), Multimodal Large Language Models (MLLMs) have made significant improvements. Many works incorporate additional module inputs on LLMs, effectively bridging the gap between visual and language. BLIP-2 (Li et al., 2023c) encodes images using ViT (Dosovitskiy et al., 2020) and employs a Q-Former to map visual features into the language space. LLaVA (Liu et al., 2023b;a; 2024a; Li et al., 2024a) utilizes an MLP as the connector between the visual encoder and the LLM backbone. Similarly, mPLUG-Owl2 (Ye et al., 2023) employs a modality-adaptive module to facilitate the collaboration between visual and language modalities by mapping them into a unified representation space. Subsequent works (Wang et al., 2023; Lu et al., 2024; Chen et al., 2024c; Young et al., 2024; Laurençon et al., 2024; GLM et al., 2024; Yao et al., 2024; Anthropic, 2024; Wang et al., 2024)further enhance MLLMs by designing novel modules for more sufficient modality alignment.

### 2.2 IMAGE IMPLICATION UNDERSTANDING

Image implication understanding represents a more complex and challenging task than conventional image understanding. This advanced cognitive process necessitates multi-hop reasoning ability and sophisticated theory of mind (ToM), capabilities that are intrinsic to human cognition (Desai et al., 2022; Hessel et al., 2023; Yang et al., 2024; Zhong et al., 2024; Strachan et al., 2024; Street et al., 2024; Horvitz et al., 2024). II-Bench (Liu et al., 2024b) is the first benchmark specifically designed to evaluate MLLMs in both image understanding and reasoning through English image implication.

## 3 THE CII-BENCH

### 3.1 OVERVIEW OF CII-BENCH

We present the **C**hinese **I**mage **I**mplication Understanding **Bench**mark (CII-Bench), a novel benchmark designed to assess the perceptual, reasoning, and comprehension abilities of MLLMs in the context of Chinese imagery. This benchmark includes a diverse range of visual content such as traditional Chinese traditional artworks, comics, posters, and Chinese Internet memes, all rich in visual information and cultural significance. The main goal of CII-Bench is to evaluate if current MLLMs can leverage their understanding and knowledge of Chinese culture to accurately interpret the deeper implications and abstract information within these images.

CII-Bench comprises 698 images across various categories, with detailed classification and domain statistics provided in Appendix A. These images are manually collected and annotated by 30 undergraduate students from different disciplines and institutions, sourced from several well-known image websites. Each image is paired with 1 to 3 multiple-choice questions, each offering six options with only one correct answer. One fixed question asks, "What is the implication in this image?" Additional questions for the same image probe different levels of understanding, such as overarching interpretation and nuanced details. The benchmark includes 800 multiple-choice questions, with 765 for the test set and 35 for developing and validating few-shot tasks. Figure 3 provides representative examples from CII-Bench.

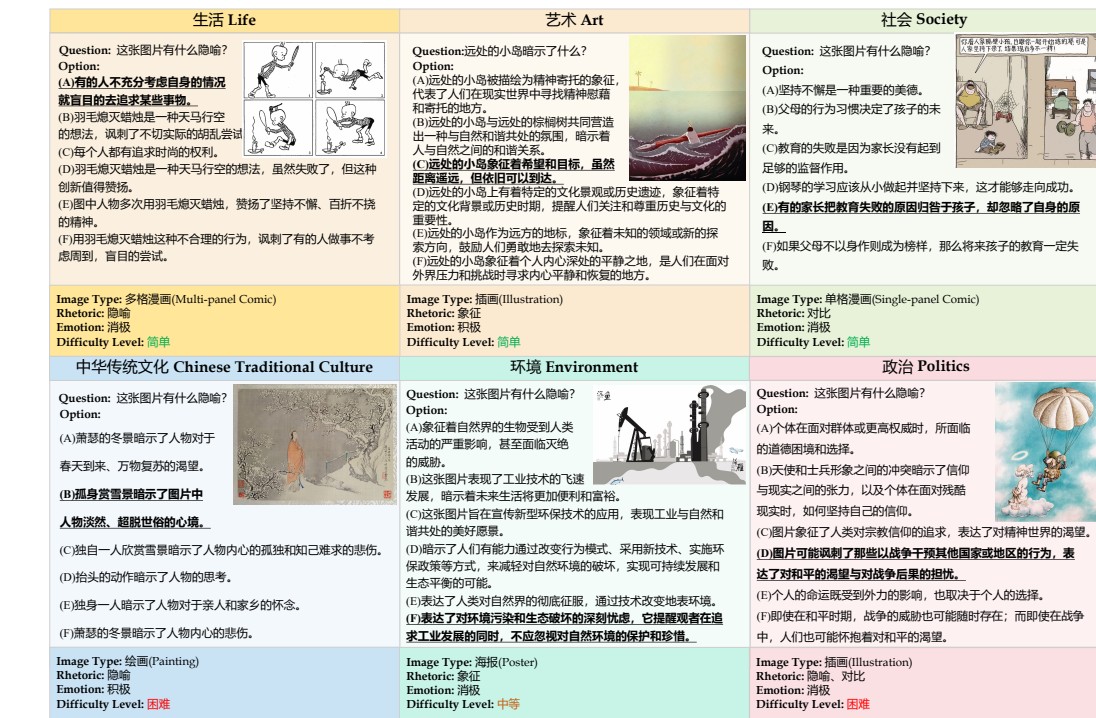

Figure 3: CII-Bench examples sampled from each domain. The English version in Appendix B.

## 3.2 DATA CURATION PROCESS

### 3.2.1 DATA COLLECTION

We collect 17,695 raw images from various renowned illustration websites, ensuring a sufficiently extensive raw dataset. Our collectors are well instructed to adhere to copyright and license regulations, avoiding data from sites prohibiting copy and redistribution. For detailed information on the specific websites from which we collect images, please refer to Appendix D.

### 3.2.2 DATA FILTRATION

After collecting the raw images, we meticulously design a three-stage data filtering process: In the first stage, we focus on image deduplication. We utilize image similarity algorithms for pixel-level comparison to eliminate duplicates and preserve dataset uniqueness; In the second stage, we regulate text prevalence in images. Optical Character Recognition (OCR) technology identifies textual areas and disqualifies images exceeding set text-area ratios, maintaining a visual-centric dataset; In the third stage, images undergo rigorous visual inspection, discarding those without metaphorical depth based on strict criteria. This process refines the dataset, rejecting over 95% of initial images and securing under 1,000 high-quality ones.

### 3.2.3 DATA ANNOTATION

The annotation process for the benchmark was meticulously designed through several steps to ensure rigor and precision as following. The detailed annotation protocol can be found in Appendix D.

**Preparation and Consistency Check:** Before formal annotation, annotators first acquaint themselves with standard templates and guidelines. A pre-annotation round on a shared image batch ensures uniform standard understanding, with discrepancies resolved through discussion.

**Multiple Rounds of Annotation and Cross-Validation:** To reduce bias, each image receives annotations from two different annotators. Cross-validation follows, with a third-party review for significant discrepancies, guaranteeing accuracy.

**Refinement of Annotation Content:** Annotators annotate each image's difficulty, type, emotional label, domain, and rhetorical devices based on specific criteria, ensuring consistency and comparability. They also craft 1 to 3 refined questions per image, each with one correct answer among five distractor options, including the default question, "What is the implication in this image?"

**Context Analysis:** During the annotation process, annotators assess the image's cultural and background significance, especially for implications and rhetorical devices, consulting relevant materials for accuracy.

**Post-Annotation Review:** Upon completion, annotations undergo a thorough quality review for any oversight, errors, or inconsistencies. Based on the evaluation results, feedback is provided to the annotators, with re-annotations as necessary to maintain data quality.

## 3.3 DATASET STATISTICS

CII-Bench comprises 698 images, each accompanied by 1 to 3 multiple-choice questions, totaling 800 questions. We randomly select 35 of these questions to construct a few-shot development set and validation set. On average, each question is approximately 11 characters long, while each option has an average length of 28 characters. Additionally, each image is supplemented with a manually written description by the annotators, which provides a detailed explanation of the image's content, nuances, and the human interpretation of its deep implication.

CII-Bench covers images across six distinct domains: Life, Art, Society, Politics, Environment, and Chinese Traditional Culture. The types of images are diverse, including Illustration, Meme, Poster, Single-panel Comic, Multi-panel Comic, and Painting. Based on human understanding, these images are categorized into three levels of difficulty: Easy, Medium, and Hard. Moreover, the images are classified according to the emotional information they convey: Positive, Neutral, or Negative. Each image is also manually annotated with the rhetorical devices employed, including Metaphor, Exaggeration, Symbolism, Visual Dislocation, Antithesis, Analogy, Personification, and Contrast. Detailed statistical information is provided in Appendix A.

## 4 EXPERIMENT

We conduct systematic experiments on both open-source and closed-source MLLMs using CII-Bench. For each model, we employ eight different configurations: None (zero-shot), 1-shot, 2-shot, 3-shot, CoT, Domain, Emotion, and Rhetoric. "None" represents the use of a standard prompt without any additional information. "Emotion" indicates the inclusion of information related to the emotional polarity of the image (e.g., positive, negative) in the prompt, "Domain" involves adding information about the image's domain (e.g., life, art), and "Rhetoric" refers to including details about the rhetorical devices used in the image (e.g., metaphor, contrast) in the prompt. Additionally, to verify the necessity of images in problem-solving, we select a portion of LLMs to complete tasks without image input. For consistency across all MLLMs and LLMs, we use identical prompts and experiment setup, with specific details available in Appendix E.

### 4.1 BASELINES

**MLLMs.** To comprehensively evaluate CII-Bench, we carefully select a diverse range of MLLMs, encompassing both open-source and closed-source models, with the aim of covering a wide spectrum of model characteristics and scales. These models span parameter sizes from 7B to 100B, ensuring that models of varying complexity and capability are thoroughly assessed. In selecting the models, we focus on the following key aspects: 1) model diversity, 2) Open-Source vs. Closed-Source models, and 3) model parameter scaling law.

**LLMs.** To verify the critical role of images in answering questions, we specifically design an experiment in which some LLMs participate in the task without any image input. The purpose of this experiment is to assess whether these models can accurately understand the questions and make correct choices in the absence of image information, thereby further demonstrating the importance of images in the comprehension and problem-solving process. We select DeepSeek-67B, LLaMA-3-8B, and Qwen2-7b as the LLMs used in this experiment.

| Model | Overall (800) | Life (216) | Art (123) | Society (157) | Politics (21) | Env. (51) | CTC (130) | Positive (220) | Negative (247) | Neutral (231) |
|---|---|---|---|---|---|---|---|---|---|---|
| *Open-source Models* | | | | | | | | | | |
| Qwen-VL-Chat | 34.3 | 27.9 | 34.7 | 32.5 | 45.8 | 55.2 | 36.5 | 34.0 | 35.1 | 33.6 |
| idefics2-8b | 36.3 | 25.0 | 46.3 | 38.1 | 41.7 | 56.9 | 32.9 | 32.8 | 39.1 | 36.4 |
| MiniCPM-Llama3-2.5 | 40.4 | 36.3 | 45.6 | 37.1 | 50.0 | 51.7 | 40.2 | 43.2 | 37.0 | 41.3 |
| CogVLM2-Llama3-Chinese-Chat | 43.4 | 37.1 | 48.3 | 42.3 | 54.2 | 63.8 | 40.2 | 40.3 | 45.7 | 43.8 |
| MiniCPM-v2.6 | 45.0 | 37.5 | 47.6 | 49.5 | 58.3 | 55.2 | 42.3 | 45.6 | 44.6 | 44.9 |
| LLaVA-1.6-34B | 46.0 | 40.8 | 55.1 | 42.8 | 45.8 | 62.1 | 43.1 | 44.4 | 48.2 | 45.2 |
| LLaVA-1.6-72B | 48.0 | 43.8 | 48.3 | 49.5 | 70.8 | 60.3 | 43.8 | 41.5 | 52.5 | 49.2 |
| Qwen2-VL-7B | 49.6 | 42.5 | 51.7 | 54.1 | 62.5 | 65.5 | 44.5 | 50.2 | 47.5 | 51.2 |
| GLM-4V-9b | 50.3 | 46.7 | 48.3 | 53.6 | 54.2 | 62.1 | 48.2 | 51.9 | 52.9 | 46.3 |
| InternVL2-Llama3-76B | 52.9 | 50.8 | 53.7 | 51.0 | 58.3 | 67.2 | 51.1 | 54.8 | 51.8 | 52.3 |
| InternVL2-8B | 53.1 | 49.2 | 53.1 | 55.7 | 62.5 | 63.8 | 50.4 | 50.6 | 53.3 | 55.1 |
| InternVL2-40B | 57.9 | 55.8 | 55.1 | 61.9 | 62.5 | 70.7 | 52.6 | 54.4 | 58.0 | 60.8 |
| Qwen2-VL-72B | **64.4** | **61.7** | **61.2** | **68.0** | **79.2** | **75.9** | **59.9** | **62.7** | **63.8** | **66.4** |
| *Closed-source Models* | | | | | | | | | | |
| GPT-4o | 54.1 | 54.1 | 55.8 | 52.1 | 50.0 | 63.8 | 51.8 | 51.9 | 56.2 | 54.1 |
| Claude-3.5-Sonnet | 54.1 | 52.1 | 61.9 | 52.6 | 62.5 | 46.6 | 53.3 | 52.7 | 56.5 | 53.0 |
| Qwen-VL-MAX | 56.9 | 53.3 | 59.2 | 58.8 | 62.5 | 67.2 | 52.6 | 53.9 | 58.3 | 58.0 |
| Gemini-1.5 Pro | 60.1 | 60.0 | 63.3 | 62.4 | 70.8 | 62.1 | 51.1 | 54.8 | 65.6 | 59.4 |
| GLM-4V | **60.9** | 55.0 | 59.9 | **66.5** | 66.7 | **79.3** | 55.5 | **58.5** | 64.5 | 59.4 |
| *Text-Only Models* | | | | | | | | | | |
| Llama-3-8B-Instruct | 21.7 | 22.2 | 26.9 | 18.6 | 25.0 | 27.8 | 20.4 | 21.2 | 24.4 | 19.5 |
| DeepSeek-67B-Chat | 27.1 | 26.6 | 32.7 | 30.9 | 20.0 | 35.2 | 18.2 | 25.7 | 22.2 | 33.2 |
| Qwen2-7B-Instruct | **32.5** | **33.2** | **34.6** | 30.9 | **35.0** | **40.7** | **28.5** | **33.6** | 30.4 | **33.6** |
| *Humans* | | | | | | | | | | |
| Human_avg | 78.2 | 81.0 | 67.7 | 82.7 | 87.7 | 84.0 | 65.9 | 77.9 | 75.2 | 81.6 |
| Human_best | **81.0** | **83.2** | **73.6** | **87.2** | **89.5** | **86.0** | **66.7** | **78.2** | **78.8** | **83.3** |

Table 1: Overall results of different MLLMs, LLMs and humans on different domains and emotions. The best-performing model in each category is **in-bold**, and the second best is underlined.

**Evaluation.** We use accuracy as the primary evaluation metric, multi-choice format questions and answer extraction method, which are widely used in previous benchmarks such as Helleswag (Zellers et al., 2019), MMMU (Yue et al., 2023), CMMMU (Zhang et al., 2024b), MMLU (Li et al., 2024b) and so on. Since CII-Bench is entirely composed of multiple-choice questions, the evaluation process only requires extracting the selected option from the model's response, which simplifies the complexity of rule design. It is important to note that when models use chain-of-thought (CoT) prompts, the responses may include intermediate steps. Therefore, the evaluation rules must be sufficiently robust, or the model's output must follow a fixed format. If the selected option cannot be extracted from the model's response, the model is considered to have answered the question incorrectly. For the detailed statistics of the model output, please see Appendix G. For reference, we also select three Chinese PhD students to evaluate human performance on CII-Bench.

## 4.2 Main Results

In this section, we conduct a comprehensive comparison of the performance of various MLLMs, LLMs, and humans on CII-Bench. Detailed results across different domains and emotional dimensions are presented in Table 1, while different image types, difficulty levels, and rhetoric can be found in Appendix F. The main experimental results and findings are summarized as follows:

### 4.2.1 Natural Challenges of CII-Bench

This benchmark presents a significant challenge for current models. Notably, despite GPT-4o being an advanced model, its accuracy is only 54.1%, indicating substantial room for improvement. This reflects the rigorous and demanding nature of the benchmark. Further analysis reveals that most models perform worst in the domain of Chinese traditional culture, highlighting a significant deficiency in their understanding of Chinese cultural nuances. It is also noteworthy that human performance in this domain is not ideal, as questions related to Chinese traditional culture often require deep cultural knowledge. The lack of this knowledge base poses difficulties for both models and humans when dealing with Chinese cultural content. In addition, text-only models like DeepSeek-67B-Chat only get 27.1% accuracy, which shows that most of the questions in CII-Bench require

| Model | None | CoT | Domain | Emotion | Rhetoric |
|---|---|---|---|---|---|
| *Open-source Models* | | | | | |
| Qwen-VL-Chat | 34.3 | 34.0 | 32.1 | 35.0 | 33.4 |
| idefics2-8b | 36.3 | 33.3 | 37.5 | 38.6 | 37.4 |
| MiniCPM-Llama3-2.5 | 40.4 | 35.8 | 41.1 | 39.0 | 34.8 |
| CogVLM2-Llama3-Chinese-Chat | 43.4 | 42.6 | 43.5 | 44.0 | 43.4 |
| MiniCPM-v2.6 | 45.0 | 38.9 | 44.4 | 45.4 | 45.4 |
| LLaVA-1.6-34B | 46.0 | 44.5 | 46.4 | 47.1 | 45.4 |
| LLaVA-1.6-72B | 48.0 | 45.3 | 47.3 | 48.6 | 45.4 |
| Qwen2-VL-7B | 49.6 | 50.0 | 51.0 | 50.8 | 49.3 |
| GLM-4V-9b | 50.3 | 49.1 | 49.9 | 51.1 | 49.5 |
| InternVL2-Llama3-76B | 52.9 | 52.6 | 54.1 | 52.8 | 53.5 |
| InternVL2-8B | 53.1 | 47.9 | 53.5 | 56.3 | 53.8 |
| InternVL2-40B | 57.9 | 57.6 | 57.1 | 60.0 | 57.9 |
| Qwen2-VL-72B | **64.4** | **62.1** | **66.0** | **64.3** | **63.0** |
| *Closed-source Models* | | | | | |
| GPT-4o | 54.1 | **54.9** | 55.4 | 54.9 | 51.9 |
| Claude-3.5-Sonnet | 54.1 | 51.6 | 56.4 | 53.5 | 54.9 |
| Qwen-VL-MAX | 56.9 | 54.0 | 59.1 | 59.9 | 54.8 |
| Gemini-1.5 Pro | 60.1 | 54.1 | 59.0 | 57.9 | 55.6 |
| GLM-4V | **60.9** | 48.8 | **60.4** | **60.6** | **58.8** |

Table 2: Overall results of different prompts on CII-Bench. The label (*Emotion, Domain, Rhetoric*) means providing corresponding information for the images in the prompt. The best-performing model in each category is **in-bold**, and the second best is underlined.

image information to be answered correctly, proving that CII-Bench is highly dependent on visual content (Chen et al., 2024a).

### 4.2.2 GAP BETWEEN HUMANS AND MLLMS

The results indicate a significant gap between human performance and multimodal large models (MLLMs) on CII-Bench. Human participants achieved an average accuracy of 78.2%, with the highest accuracy reaching 81.0%. In contrast, the best-performing closed-source model, GLM-4V, achieved an accuracy of 60.9%, while the top open-source model, Qwen2-VL-72B, scored 64.4%. These findings highlight the substantial disparity between human abilities and even the most advanced models in understanding image implications. The highest accuracy achieved by the models is considerably lower than the average human score, indicating that multimodal large models still face significant challenges in this domain.

### 4.2.3 MODEL PERFORMANCE ACROSS DIFFERENT DOMAINS AND EMOTIONS

In terms of domain performance, our results in Table 1 indicate that the models generally perform better in the Environment and Politics domains, achieving higher accuracy. Conversely, the accuracy is lower in the Life and Society domains, proving that everyday metaphors are generally more difficult in the Chinese context. The lowest score for the Chinese Traditional Culture and Art domains, which shows that while the models generalize well in common domains, they struggle with the more abstract and logically demanding information found in Chinese Traditional Culture and Art.

From an emotional perspective, the models tend to exhibit higher accuracy when the image implications convey negative emotions, while accuracy is the lowest for images with positive emotions. This discrepancy highlights that the models' preferences do not align with those of humans, as humans are significantly more sensitive to positive implications. The performance of the model is opposite to the conclusion shown in II-Bench (Liu et al., 2024b), reflecting the obvious difference in emotional expression in the Chinese and English contexts.

### 4.2.4 ANALYSIS ON DIFFERENT PROMPT SKILLS

**Analysis of Chain-of-Thought (CoT).** In Table 2, we evaluate the impact of Chain-of-Thought (CoT) prompting on model performance. The results indicate that CoT does not significantly improve the accuracy of the models. In some cases, particularly with smaller open-source models, the

accuracy even declined when CoT was used. For example, MiniCPM-v2.6 scores 45.0% without CoT, but this drops to 38.9% with CoT; similarly, LLaVA-1.6-72B scores decrease from 48.0% to 45.3%.

Upon analyzing the models' responses, we find that those models showing a decrease in accuracy with CoT often suffer from overinterpretation, where questions that were initially answered correctly are misinterpreted after CoT is applied. Additionally, for questions that were originally answered incorrectly, CoT does not lead to significant improvements and sometimes even causes confusion, such as selecting multiple options. However, for most models, the probability of failing to extract an answer option from the response decreases after using CoT, which explains why some models show improved accuracy with CoT.

**Analysis of Different Types and Domains.** To evaluate the impact of different label information on model accuracy, we conduct an ablation study by providing relevant label information (such as emotion, domain, and rhetoric) in the prompts. The results in Table 2 show that emotion labels significantly improve model accuracy, followed by domain and rhetoric labels, both of which exhibit similar effectiveness.

This result aligns with human intuition. The answer options typically include negative, positive, and neutral choices. When the model receives emotional information, it can eliminate some irrelevant options, naturally leading to higher accuracy. In contrast, domain and rhetoric information generally do not effectively help the model eliminate options, resulting in more limited improvements. Additionally, from a model training perspective, models tend to have a more mature understanding of emotions, while specific nouns in rhetoric and domain labels are often custom-defined. During pre-training, the model may not have encountered a large number of descriptions for such specific nouns, making these labels less helpful in improving accuracy.

**Analysis of Few-shot Examples.** The results in Table 3 indicate that few-shot examples do not improve the models' accuracy. Specifically, performance declines as the number of examples increases. This decline can be attributed to the models' inferior capabilities in handling multiple images compared to single images, leading to a decrease in accuracy with a higher number of shots. Furthermore, as the number of shots increases, the input length also extends, and the models' ability to process long texts is inadequate, resulting in suboptimal performance with long contexts.

| Model | None | 1-shot | 2-shot | 3-shot |
|---|---|---|---|---|
| Qwen2-VL-7B | 49.6 | 44.1 | 39.3 | 37.5 |
| GPT-4o | 54.1 | 51.8 | 49.5 | 49.1 |
| Claude-3.5-Sonnet | 54.1 | 55.4 | 55.3 | 55.4 |
| InternVL2-40B | 57.9 | 53.0 | 47.1 | 41.9 |
| Gemini-1.5 Pro | 60.1 | 57.4 | 55.8 | 55.4 |

Table 3: Few-shot results of different models on the CII-Bench.

## 4.3 EVALUATION OF CHINESE TRADITIONAL CULTURE

The Chinese traditional culture category is a distinctive feature of the CII-Bench dataset, where MLLMs consistently score the lowest. Therefore, we need a deeper evaluation of this field to analyze the extent to which MLLM understands Chinese traditional culture. We chose to deeply analyze MLLM's understanding of Chinese traditional culture by evaluating Chinese traditional paintings.

### 4.3.1 EVALUATION METRIC

Chinese traditional painting, a cornerstone of Chinese traditional culture, encompasses a rich tapestry of styles and techniques developed over millennia. These paintings are typically categorized based on their subject matter (e.g., landscape paintings, flower-and-bird paintings, figure paintings, and New Year paintings) or their stylistic and skill (e.g., court paintings, meticulous brush paintings, freehand brush paintings, and color-and-ink paintings). Each category embodies unique characteristics that reflect China's artistic evolution and philosophical underpinnings.

To comprehensively assess MLLMs' understanding of Chinese traditional paintings, we develop a multifaceted evaluation metric. This metric is designed to probe both the surface-level information

readily apparent in the artwork and the deeper culture and history that informs its creation and interpretation. Our evaluation metric encompasses five key perspectives: ***Surface-level Information***, ***Aesthetic Characteristics***, ***Brush and Ink Skills***, ***Culture and History***, and ***Deep Implications***. For each perspective, we give its detailed description in Appendix C.

### 4.3.2 LLM-BASED CHINESE TRADITIONAL PAINTING AUTOMATIC EVALUATION

To evaluate Chinese traditional painting comprehension in MLLMs, we develop an LLM-based evaluation standard based on evaluation metrics, as illustrated in Appendix C. Our experiment utilize the CTC domain data from CII-Bench, comprising 130 Chinese traditional paintings. We employ human-written descriptions and implication interpretations as ground truth. We choose GPT-4o to generate descriptions for these images, which are subsequently scored using GPT-4o and our evaluation standard. Please see the evaluation prompt in Appendix E. To validate the model's scoring efficacy, we enlist three PhD students well-versed in Chinese metaphorical imagery to independently score the 130 paintings.

The model-human scoring consistency reached 98%, affirming the method's validity for assessing Chinese traditional painting comprehension. Table 4 presents the detailed model scores. Analysis of these results, in conjunction with our evaluation standard, reveals insights across three dimensions: overall performance, difficulty levels, and emotions. The overall score of 2.71 indicates that while MLLM is able to observe the surface-level information of paintings, it has a large gap with humans in deeply interpreting the complex cultural elements contained in Chinese traditional art. In terms of difficulty evaluation, the model is consistent with human cognition, while in terms of emotion, the model has a higher negative score, indicating that the model can identify negative implications in paintings, such as using the past to satirize the present, and not appreciating talents.

| Model | Overall | Easy | Middle | Difficult | Positive | Negative | Neutral |
|-------|---------|------|--------|-----------|----------|----------|---------|
| GPT-4o | 2.71 | 3.0 | 3.2 | 2.35 | 2.63 | 3.0 | 2.82 |

Table 4: Overall result of Chinese traditional painting.

### 4.4 ERROR ANALYSIS

To conduct a comprehensive error analysis of GPT-4o's performance (under CoT setting) on CII-Bench, we randomly select a total of 100 erroneous samples from various domains, distributed according to their proportions in the dataset. These samples are subjected to in-depth analysis by expert annotators. As illustrated in Figure 4, GPT-4o's errors can be categorized into the following types: Information Neglect, Misunderstanding of Visual Information, Over-Inference, Superficial Reasoning, and Lack of Cultural Background Knowledge. For detailed analysis of cases, please see the Appendix I.

**Information Neglect (36%):**

Complex images contain both visual and textual elements. Sole reliance on visual information makes accurate interpretation challenging due to diversity in meaning. Incorporating textual information clarifies the author's emotional intent, aiding accurate interpretation. Unfortunately, GPT-4o frequently overlooks key visual (13%) and textual (23%) information. When directly asked about these elements, we find that GPT-4o can often answer correctly, indicating two main issues: 1) Insufficient image recognition abilities, and 2) Significant shortcomings in multimodal fusion, leading to underutilization of acquired information.

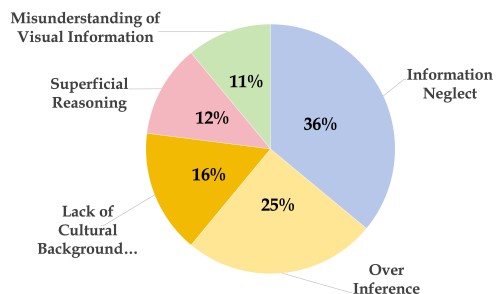

Figure 4: GPT-4o error responses distribution.

**Over-Inference (25%):**

During answer construction, distractors are included at surface and deep levels. GPT-4o often selects more exaggerated, deep-level incorrect options, ignoring narrower but correct ones, especially in Chinese memes. This suggests that GPT-4o has a preference for selecting abstract options.

**Other Errors:** The remaining errors are *Lack of Cultural Background Knowledge* (16%), *Superficial Reasoning* (12%), *Misunderstanding of Visual Information* (11%). The description of these errors can be found in Appendix H.

## 5 DISCUSSION

### 5.1 INTERPRETABILITY ANALYSIS OF CHINESE IMAGE IMPLICATIONS

The essence of Chinese image implications is deeply rooted in deep cultural heritage and complex contextual associations, which enables them to convey profound messages through nuanced expressions. For example, in traditional Chinese art forms such as landscape and New Year paintings, the imagery transcends mere depiction of nature or daily occurrences. Instead, it embodies emotions, philosophical insights, and societal norms through metaphorical and highly symbolic expressions. These symbols, like the pine tree, plum blossom, and crane, are not superficial meaning but are steeped in centuries of cultural tradition, representing resilience, purity, and longevity.

However, deciphering these complex messages can be challenging, particularly for those unfamiliar with the cultural and historical narratives tied to these symbols. This contrasts with English image implications, which often convey messages through more straightforward and explicit symbolism. As a result, the interpretability of Chinese image implications depends to some extent on reconstructing and resonating with the cultural context, which is what makes them unique: their meaning is not only visual but also culturally resonant, bridging across time and space.

Moreover, the interpretability of Chinese image implications has new changed in the modern era. Globalization and the surge of internet culture have intertwined foreign elements with traditional Chinese culture, birthing new symbols and implications. This intersection introduces additional layers of meaning, complicating the interpretation of traditional symbols.

### 5.2 WHY CHOOSE CHINESE TRADITIONAL PAINTINGS TO EVALUATE CHINESE TRADITIONAL CULTURE?

The imagery associated with Chinese traditional culture often embodies complex implications, encompassing customs, historical anecdotes, and legendary tales, making direct evaluation particularly challenging. Chinese traditional painting, intrinsically intertwined with Chinese traditional culture, offers a viable proxy for this assessment. The unique value of Chinese traditional painting lies in its embodiment of Chinese cultural connotations, aesthetic implications, and distinctive artistic expression. The core philosophical concepts of Confucianism, Taoism, and Buddhism, along with their humanistic essence, have consistently permeated the entire trajectory of Chinese painting history. Consequently, we have chosen to evaluate MLLMs' comprehension of Chinese traditional culture through an in-depth analysis of their understanding of Chinese traditional paintings.

## 6 CONCLUSION

The development of CII-Bench marks a significant step forward in evaluating the capabilities of multimodal large models (MLLMs) and brings us closer to achieving expert artificial general intelligence (AGI). This benchmark promotes a deeper exploration of the higher-order theory of mind in MLLMs. Experimental results indicate that current MLLMs still exhibit a significant gap compared to humans in understanding the implications of images within a Chinese context. We found that most MLLMs lack a deep knowledge base of Chinese traditional culture, leading to a superficial understanding of this cultural content. Finally, the experiments showed that incorporating image emotion hints into prompts often improves model performance, suggesting that models still struggle with emotional understanding, which in turn leads to misinterpretation of implications. We believe that CII-Bench will inspire the academic community to further develop the next generation of multimodal foundational models that move toward expert AGI.

LIMITATIONS

We acknowledge several limitations in our study. Although CII-Bench is comprehensive, subjective elements can result in varying interpretations, impacting result consistency. In addition, in order to ensure high quality and practicability, our benchmark is not particularly large. The evaluation metrics may not fully capture the advanced understanding and reasoning capabilities of AI systems. These limitations underscore the necessity for continuous refinement and expansion of our benchmarks. Future work will focus on developing and incorporating more stringent and objective test sets to enhance the reliability and validity of our benchmark.

ETHICS STATEMENT

In developing CII-Bench, we strictly adhere to ethical guidelines and legal regulations, ensuring fairness, transparency, inclusivity and respect for all stakeholders. We stress the importance of safeguarding privacy and intellectual property rights, underscoring our commitment to responsible and lawful data management. We have taken steps to anonymize any personal data to protect privacy and and have made every effort to minimize harmful or biased content. However, we recognize that biases can inadvertently arise and some information may be potentially offensive. We are committed to continuous monitoring and improvement to mitigate such biases. Furthermore, we encourage users of our dataset to employ it responsibly and to consider the ethical implications of their work, particularly in applications that may impact individuals or communities.

REPRODUCIBILITY

Our paper consists of the CII-Bench dataset and evaluation experiments, which we will describe in detail to ensure reproducibility. Regarding the dataset, specific information about the websites from which we collect images is available in Appendix D. Additionally, we provide an overview of the data curation process in **Section 3.2**. As for the experiments, we employ different prompts and experimental setups. Detailed descriptions can be found in Appendix E. We provide the CII-Bench dataset and all experiment codes with detailed descriptions in the README.md file in the **Supplementary Material**. We believe that our comprehensive descriptions significantly contribute to the reproducibility of our research.

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

# A  STATISTICS OF CII-BENCH

| Statistics | |
|---|---|
| Total Questions | 800 |
| Total Images | 698 |
| Dev : Validation : Test | 15 : 20 : 765 |
| Easy : Medium : Hard | 305 : 282 : 111 |
| Average Question Length | 10.54 |
| Average Option Length | 28.31 |
| Average Explanation Length | 121.06 |
| Metaphor | 562 |
| Exaggerate | 121 |
| Symbolism | 236 |
| Visual Dislocation | 42 |
| Antithesis | 13 |
| Analogy | 19 |
| Personification | 73 |
| Contrast | 87 |

| Statistics | |
|---|---|
| Life | 216 (30.95%) |
| Art | 123 (17.62%) |
| Society | 157 (22.49%) |
| Environment | 51 (7.31%) |
| Politics | 21 (3.01%) |
| Chinese Traditional Culture | 130 (18.62%) |
| Positive | 220 (31.52%) |
| Neutral | 247 (35.39%) |
| Negative | 231 (33.09%) |
| Illustration | 178 (25.50%) |
| Meme | 145 (20.77%) |
| Poster | 87 (12.46%) |
| Multi-panel Comic | 34 (4.87%) |
| Single-panel Comic | 143 (20.49%) |
| Painting | 119 (17.05%) |

Table 5: General statistics of CII-Bench.

Figure 5: CII-Bench specific image type and domain statistics.

## B CII-BENCH EXAMPLES OF ENGLISH VERSION

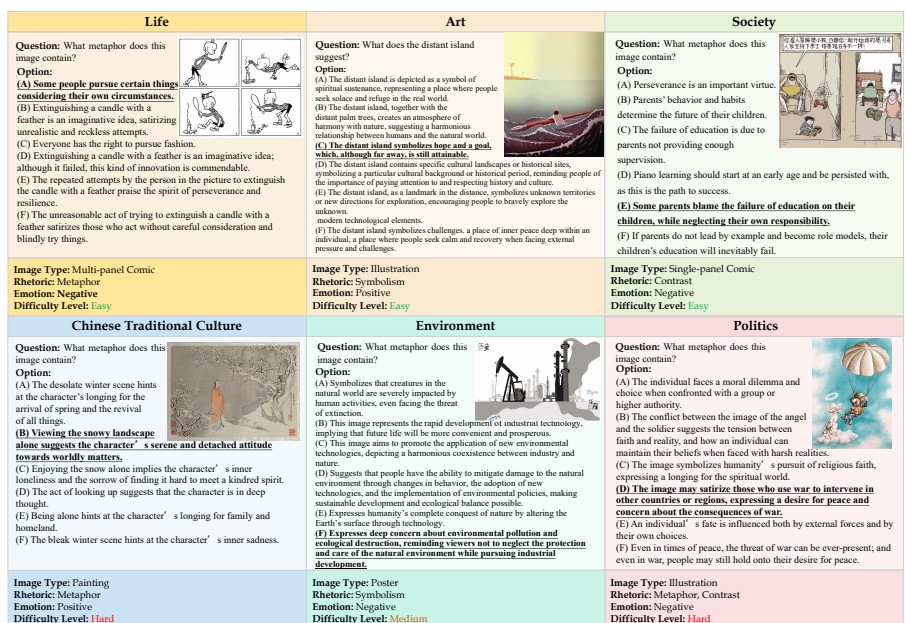

Figure 6: CII-Bench examples sampled from each domain. The pictures include life, art, society, Chinese traditional culture, environment and politics. Understanding these images and completing the corresponding questions require a certain level of comprehension.

## C EVALUATION METRIC AND STANDARD OF CHINESE TRADITIONAL CULTURE

| Evaluation Metric | Evaluation Standard |
|---|---|
| **1. Surface-level Information:**
• Identification of primary entities within the painting
• Analysis of color composition and application
• Recognition of intricate details and their significance

**2. Aesthetic Characteristics:**
• Comprehension of compositional techniques
• Recognition of the painting's artistic school or movement
• Interpretation of emotional expression through visual elements

**3. Brush and Ink Skills:**
• Identification of specific painting skills
• Analysis of expressive methods employed by the artist

**4. Culture and History:**
• Understanding of the painting's historical background
• Interpretation of the artist's intended message or philosophy
• Placement of the artwork within its appropriate historical period

**5. Deep Implications:**
• Recognition of traditional cultural elements embedded in the painting
• Interpretation of philosophical concepts, humanistic spirit, or social values conveyed
• Demonstration of a deep understanding of Chinese traditional culture as reflected in the artwork | **[1 point]:**
The description of the picture is incomplete and does not include any background information. It only mentions the most obvious elements in the picture, lacking recognition and understanding of traditional cultural elements.

**[2 points]:**
The description of the image is relatively complete, but the background information is limited. Some traditional cultural elements can be identified, but their meaning is not explained in depth. There is a lack of insight into the aesthetic characteristics or skills of the image.

**[3 points]:**
The description of the image is comprehensive and includes some background information. The traditional cultural elements can be accurately identified and briefly explained. There is a preliminary understanding of the aesthetic characteristics or skills of the picture, but there is a lack of in-depth analysis. Some historical or cultural background is mentioned, but it is not fully expanded.

**[4 points]:**
The description of the image is detailed and contains rich background information. It is able to explain the significance of traditional cultural elements in depth. It provides a relatively in-depth analysis of the aesthetic characteristics and skills of the image. It provides relevant historical and cultural background and attempts to explore the implication of the image, but the implication is misunderstood.

**[5 points]:**
The description of the image is comprehensive and accurate, and the background information is in-depth and comprehensive. It can not only accurately interpret traditional cultural elements, but also reveal their deep meaning. It has unique insights into the aesthetic characteristics and skills of the image. It deeply expounds on the relevant historical and cultural background, and can see through the philosophical thoughts, humanistic spirit or social values contained in the image, showing a deep understanding of Chinese traditional culture. |

Figure 7: Evaluation metric and evaluation standard of Chinese traditional painting.

# D DATA ANNOTATION PROTOCOL

This document outlines a comprehensive protocol for annotating a dataset consisting of questions that explore the metaphorical implications of images.

## D.1 DATA COLLECTION

Some websites from which we collect data are as follows:

- `https://fabiaoqing.com/biaoqing/lists/page.html`
- `https://www.sohu.com/a/282205200_439969`
- `https://www.sohu.com/a/300233985_616741`
- `https://www.zcool.com.cn/u/746800`
- `https://www.shencaitang.com/news/1940`
- `https://www.dpm.org.cn/collection/paints.html`
- `https://www.zuomeme.com/wangyou/all`

## D.2 GENERAL GUIDELINES

**General Principles:**

- Annotations should be accurate and consistent.
- All questions, options and explanations should be written in Chinese.
- Any images without metaphorical implications should be discarded.

**Specific Instructions:**

- Each image needs to be categorized as one of the following image types: single-panel comic, multi-panel comic, poster, meme, illustration or painting.
- Each image needs to be categorized as one of the following difficulty levels from a human understanding perspective: easy, middle, or hard.
- Each image needs to be categorized as one of the following domains: life, art, society, politics, environment or Chinese traditional culture.
- Each image needs to be categorized as one of the following emotions: positive, neutral or negative.
- Each image needs to be categorized as one or more of the following rhetoric: metaphor, exaggerate, symbolism, contrast, visual dislocation, antithesis, analogy, personification or others.
- Each image needs a human explanation and implication description.
- Each image needs 1-3 questions about the fine-grained metaphorical implications of the image, each with one correct answer and five distractor options.

## D.3 DATA QUALITY ASSURANCE

To further ensure the quality and reliability of the data, the annotated datasets were double-checked and cross-validated. Each question was manually validated by at least five annotators. Any inconsistencies or misinterpretations found were thoroughly examined and resolved by consensus of the annotation team, thus improving the reliability of the dataset while ensuring consistency of the annotations. In total, we conducted five rounds of data quality checks to ensure data quality and ultimately obtain CII-Bench.

### D.4 Ethical Considerations

**Copyright and Licensing.** It is essential to strictly follow all copyright and licensing regulations. Data from sources that do not permit copying or redistribution will be explicitly excluded.

**Data Privacy.** Adherence to privacy laws and ethical standards in data handling is crucial. Annotators must avoid collecting questions that contain any personal information.

# E  EXPERIMENT SETUP

In experiments, we set the model temperature as 0, and all experiments are conducted on Nvidia A800 GPUs. The prompts of different settings are as follows Figure 8 to Figure 11. Particularly, the evaluation prompt of Chinese traditional painting is Figure 12.

---

**Evaluation Prompt: Direct**

请根据提供的图片尝试回答下面的单选题。直接回答正确选项，不要包含额外的解释。请使用以下格式："答案：**$LETTER**"，其中 **$LETTER**是你认为正确答案的字母。

**{question}**
**{options}**

答案：

---

Figure 8: The prompt used in direct output setting.

---

**Evaluation Prompt: Keywords**

请根据提供的图片尝试回答下面的单选题。请使用以下格式："答案：**$LETTER**"，其中**$LETTER**是你认为正确答案的字母。

关键词：**{key_words}**

**{question}**
**{options}**

答案：

---

Figure 9: The prompt used in keyword setting.

---

**Evaluation Prompt: CoT**

请尝试根据提供的图片回答以下单选题。让我们逐一思考每个选项，逐步分析。你回答的最后一行应该用以下格式："答案：**$LETTER**"，其中**$LETTER**是你认为正确答案的字母。

**{question}**
**{options}**

---

Figure 10: The prompt used in CoT setting.



**Evaluation Prompt: Few-Shot**

请根据提供的图片尝试回答下面的单选题。直接回答正确选项，不要包含额外的解释。请使用以下格式："答案：**$LETTER**"，其中**$LETTER**是你认为正确答案的字母。

**{question 1}**
**{options 1}**

答案：**{answer 1}**

**(one or two more examples)**

请根据提供的图片尝试回答下面的单选题。直接回答正确选项，不要包含额外的解释。请使用以下格式："答案：**$LETTER**"，其中**$LETTER**是你认为正确答案的字母。

**{question}**
**{options}**

答案：



Figure 11: The prompt used in Few-Shot setting.



**# Role**
You are an impartial judge, familiar with Chinese traditional culture and traditional paintings.

**## Attention**
You are responsible for evaluating the quality of the descriptions provided by the model for traditional Chinese paintings. Your evaluation should refer to the human answer and score based on the Evaluation Standard.

**## Evaluation Standard**
**- [1 point]:**
The description of the picture is incomplete and does not include any background information. It only mentions the most obvious elements in the picture, lacking recognition and understanding of traditional cultural elements.
**- [2 points]:**
The description of the image is relatively complete, but the background information is limited. Some traditional cultural elements can be identified, but their meaning is not explained in depth. There is a lack of insight into the aesthetic characteristics or skills of the image.
**- [3 points]:**
The description of the image is comprehensive and includes some background information. The traditional cultural elements can be accurately identified and briefly explained. There is a preliminary understanding of the aesthetic characteristics or skills of the picture, but there is a lack of in-depth analysis. Some historical or cultural background is mentioned, but it is not fully expanded.
**- [4 points]:**
The description of the image is detailed and contains rich background information. It is able to explain the significance of traditional cultural elements in depth. It provides a relatively in-depth analysis of the aesthetic characteristics and skills of the image. It provides relevant historical and cultural background and attempts to explore the implication of the image, but the implication is misunderstood.
**- [5 points]:**
The description of the image is comprehensive and accurate, and the background information is in-depth and comprehensive. It can not only accurately interpret traditional cultural elements, but also reveal their deep meaning. It has unique insights into the aesthetic characteristics and skills of the image. It deeply expounds on the relevant historical and cultural background, and can see through the philosophical thoughts, humanistic spirit or social values contained in the image, showing a deep understanding of Chinese traditional culture.

**## Constraints**
- Avoid any position biases and be as objective as possible
- Do not allow the length of the descriptions to influence your evaluation

**## Workflow**
Output your final verdict by strictly following this format: "[ratings]".



Figure 12: The prompt of Chinese traditional painting evaluation.

## F  RESULTS ON DIFFERENT TYPES, DIFFICULTIES AND RHETORIC

In this section, we report the performance of different MLLMs and humans on different types of images, levels of difficulty, and rhetoric types.

| Model | Overall | Illus. | Paint. | Poster | Single-C. | Multi-C. | Meme |
|---|---|---|---|---|---|---|---|
| *Open-source Models* | | | | | | | |
| Qwen-VL-Chat | 34.3 | 33.5 | 36.8 | 45.1 | 35.2 | 23.7 | 27.5 |
| idefics2-8b | 36.3 | 44.0 | 32.8 | 45.1 | 35.2 | 23.7 | 24.8 |
| MiniCPM-Llama3-2.5 | 40.4 | 39.5 | 38.4 | 49.0 | 42.6 | 34.2 | 37.3 |
| CogVLM2-Llama3-Chinese-Chat | 43.4 | 45.0 | 39.2 | 52.9 | 45.5 | 23.7 | 39.2 |
| MiniCPM-v2.6 | 45.0 | 44.0 | 40.8 | 53.9 | 51.1 | 36.8 | 39.2 |
| LLaVA-1.6-34B | 46.0 | 50.0 | 44.0 | 48.0 | 47.7 | 29.0 | 42.5 |
| LLaVA-1.6-72B | 48.0 | 50.9 | 44.0 | 43.1 | 56.8 | 39.5 | 43.1 |
| Qwen2-VL-7B | 49.6 | 47.7 | 43.2 | 0.8 | 58.0 | 31.6 | 46.4 |
| GLM-4V-9b | 50.3 | 46.8 | 47.2 | 55.9 | 59.7 | 42.1 | 47.1 |
| InternVL2-Llama3-76B | 52.9 | 48.2 | 50.4 | 59.8 | 62.5 | 39.5 | 49.7 |
| InternVL2-8B | 53.1 | 48.2 | 48.0 | 56.9 | 64.8 | **52.6** | 51.0 |
| InternVL2-40B | 57.9 | 53.7 | 51.2 | 56.9 | 68.2 | 50.0 | 59.5 |
| Qwen2-VL-72B | **64.4** | **61.5** | **59.2** | **68.6** | **70.5** | 47.4 | **67.3** |
| *Closed-source Models* | | | | | | | |
| GPT-4o | 54.1 | 54.1 | 50.4 | 56.9 | 54.6 | 47.4 | 57.5 |
| Claude-3.5-Sonnet | 54.1 | 55.1 | **54.4** | 47.1 | 55.1 | 50.0 | 57.5 |
| Qwen-VL-MAX | 56.9 | 57.3 | 51.2 | 60.8 | 62.5 | 39.5 | 56.2 |
| Gemini-1.5 Pro | 60.1 | **64.7** | 50.4 | 52.0 | 66.5 | **52.6** | **62.1** |
| GLM-4V | **60.9** | 59.6 | **54.4** | **67.7** | **70.5** | 44.7 | 57.5 |
| *Humans* | | | | | | | |
| Human_avg | 78.2 | 71.5 | 65.6 | 75.2 | 79.8 | 74.5 | 83.6 |
| Human_best | **81.0** | **76.9** | **66.1** | **78.6** | **81.7** | **78.4** | **85.0** |

Table 6: Overall results of different MLLMs on different image types. The best-performing model in each category is **in-bold**, and the second best is underlined. For brevity, Illus. refers to Illustration, Paint. refers to Painting, Single-C. refers to Single-panel Comic, Multi-C. refers to Multi-panel Comic.

| Model | Overall | Easy | Medium | Hard |
|---|---|---|---|---|
| *Open-source Models* | | | | |
| Qwen-VL-Chat | 34.3 | 36.3 | 33.5 | 30.3 |
| idefics2-8b | 36.3 | 35.4 | 39.3 | 30.3 |
| MiniCPM-Llama3-2.5 | 40.4 | 43.1 | 39.3 | 35.3 |
| CogVLM2-Llama3-Chinese-Chat | 43.4 | 46.3 | 39.9 | 44.3 |
| MiniCPM-v2.6 | 45.0 | 47.1 | 44.2 | 41.0 |
| LLaVA-1.6-34B | 46.0 | 44.9 | 47.0 | 46.7 |
| LLaVA-1.6-72B | 48.0 | 50.0 | 47.0 | 45.1 |
| Qwen2-VL-7B | 49.6 | 52.6 | 47.9 | 45.9 |
| GLM-4V-9b | 50.3 | 52.6 | 49.1 | 46.7 |
| InternVL2-Llama3-76B | 52.9 | 57.4 | 49.7 | 48.4 |
| InternVL2-8B | 53.1 | 57.7 | 49.4 | 50.0 |
| InternVL2-40B | 57.9 | 62.3 | 55.5 | 51.6 |
| Qwen2-VL-72B | **64.4** | **68.9** | **63.1** | **54.9** |
| *Closed-source Models* | | | | |
| GPT-4o | 54.1 | 56.0 | 54.9 | 46.7 |
| Claude-3.5-Sonnet | 54.1 | 55.1 | 52.4 | 55.7 |
| Qwen-VL-MAX | 56.9 | 57.4 | 56.7 | 55.7 |
| Gemini-1.5 Pro | 60.1 | 61.1 | **61.3** | 54.1 |
| GLM-4V | **60.9** | **62.9** | 59.2 | **59.8** |
| *Humans* | | | | |
| Human_avg | 78.2 | 82.5 | 76.1 | 70.9 |
| Human_best | **81.0** | **84.0** | **78.9** | **71.8** |

Table 7: Overall results of different MLLMs on various difficulty levels. The best-performing model in each category is **in-bold**, and the second best is underlined. The numbers in parentheses indicate the number of samples in each category.

| Model | Overall | Meta. | Exag. | Symb. | Contrast | VisD. | Pers. | Anal. | Anti. |
|---|---|---|---|---|---|---|---|---|---|
| *Open-source Models* | | | | | | | | | |
| Qwen-VL-Chat | 34.3 | 31.8 | 38.9 | 38.4 | 41.0 | 37.0 | 34.2 | 28.6 | 30.8 |
| idefics2-8b | 36.3 | 35.2 | 32.6 | 35.6 | 41.9 | 30.4 | 26.6 | 23.8 | 38.5 |
| MiniCPM-Llama3-2.5 | 40.4 | 38.5 | 42.4 | 40.2 | 38.1 | 34.8 | 44.3 | 33.3 | 38.5 |
| CogVLM2-Llama3-Chinese-Chat | 43.4 | 42.2 | 46.5 | 42.7 | 44.8 | 50.0 | 44.3 | 52.4 | 38.5 |
| MiniCPM-v2.6 | 45.0 | 41.7 | 48.6 | 43.4 | 41.0 | 45.7 | 45.6 | 38.1 | **53.9** |
| LLaVA-1.6-34B | 46.0 | 45.1 | 47.9 | 45.9 | 41.0 | 45.7 | 44.3 | 42.9 | 30.8 |
| LLaVA-1.6-72B | 48.0 | 46.1 | 54.2 | 48.0 | 49.5 | 47.8 | 46.8 | 47.6 | 38.5 |
| Qwen2-VL-7B | 49.6 | 47.6 | 52.1 | 48.4 | 49.5 | 56.5 | 51.9 | 47.6 | **53.9** |
| GLM-4V-9b | 50.3 | 48.7 | 56.3 | 51.3 | 52.4 | 50.0 | 50.6 | 57.1 | 30.8 |
| InternVL2-Llama3-76B | 52.9 | 51.5 | 59.7 | 51.3 | 51.4 | 52.2 | 55.7 | 52.4 | 46.2 |
| InternVL2-8B | 53.1 | 51.0 | 54.9 | 55.2 | 47.6 | 54.4 | 57.0 | 47.6 | 46.2 |
| InternVL2-40B | 57.9 | 55.8 | 63.2 | 56.6 | 55.2 | 54.4 | **69.6** | **71.4** | 46.2 |
| Qwen2-VL-72B | **64.4** | **62.5** | **70.1** | **65.8** | **63.8** | **73.9** | 67.1 | 66.7 | **53.9** |
| *Closed-source Models* | | | | | | | | | |
| GPT-4o | 54.1 | 52.6 | 54.9 | 51.6 | 51.4 | 60.9 | 55.7 | 52.4 | 38.5 |
| Claude-3.5-Sonnet | 54.1 | 52.1 | 54.9 | 56.6 | 47.6 | 50.0 | 54.4 | 57.1 | 38.5 |
| Qwen-VL-MAX | 56.9 | 54.7 | 60.4 | 58.7 | 52.4 | 58.7 | 55.7 | 57.1 | 46.2 |
| Gemini-1.5 Pro | 60.1 | 59.5 | 64.6 | 60.1 | **61.9** | 47.8 | 55.7 | **81.0** | **53.9** |
| GLM-4V | **60.9** | **60.2** | **65.3** | **63.4** | 57.1 | **65.2** | **60.8** | 66.7 | 46.2 |
| *Humans* | | | | | | | | | |
| Human_avg | 78.2 | 76.0 | 82.8 | 74.1 | 70.4 | 73.9 | 72.9 | 90.0 | 52.8 |
| Human_best | **81.0** | **77.0** | **85.2** | **76.5** | **75.7** | **75.6** | **74.7** | **95.0** | **66.7** |

Table 8: Overall results of different MLLMs and humans on different rhetoric. The best-performing model in each category is **in-bold**, and the second best is underlined. For brevity, Meta. refers to Metaphor, Exag. refers to Exaggerate, Symb. refers to Symbolism, VisD. refers to Visual Dislocation, Anti. refers to Antithesis, Anal. refers to Analogy, Pers. refers to Personification

# G    ADDITIONAL DETAILS OF RESULTS

We do detailed statistics of the model output. The results are shown in Table 9 to 12. *Miss* is mainly caused by two situations, one is that the model does not give an answer, and the other is the regex is not matched. The *Miss* rate of most models is controlled below an acceptable ratio. In the *CoT* setting, some models do not follow instructions well and do not provide the expected letters as answer, which cannot be matched and will be considered a *Miss*.

| Mode | Metric | InternVL2-40B | InternVL2-8B | InternVL2-Llama3-76B | MiniCPM-Llama3-2.5 | MiniCPM-v2.6 |
|------|--------|---------------|--------------|----------------------|--------------------|--------------|
| CoT | Acc | 57.6 | 47.9 | 52.6 | 35.8 | 39.3 |
| | Error | 0.0 | 0.0 | 0.0 | 0.0 | 0.0 |
| | Miss | 0.0 | 0.0 | 0.0 | 8.1 | 0.0 |
| Domain | Acc | 57.1 | 53.5 | 54.1 | 41.1 | 44.4 |
| | Error | 0.0 | 0.0 | 0.0 | 0.0 | 0.0 |
| | Miss | 0.0 | 0.0 | 0.0 | 5.9 | 0.0 |
| Emotion | Acc | 60.0 | 56.3 | 52.8 | 39.0 | 45.4 |
| | Error | 0.0 | 0.0 | 0.0 | 0.0 | 0.0 |
| | Miss | 0.0 | 0.0 | 0.0 | 8.4 | 0.0 |
| None | Acc | 57.9 | 53.1 | 52.9 | 40.4 | 45.0 |
| | Error | 0.0 | 0.0 | 0.0 | 0.0 | 0.0 |
| | Miss | 0.0 | 0.0 | 0.0 | 0.4 | 0.0 |
| Rhetoric | Acc | 57.9 | 53.8 | 53.5 | 34.8 | 45.4 |
| | Error | 0.0 | 0.0 | 0.0 | 0.0 | 0.0 |
| | Miss | 0.0 | 0.0 | 0.0 | 10.4 | 0.0 |

Table 9: Accuracy, Error and Miss rate of different models under different settings.(1/4)

| Mode | Metric | Qwen-VL-Chat | Qwen2-VL-72B | Qwen2-VL-7B | CogVLM2-Llama3-Chinese-Chat |
|------|--------|--------------|--------------|-------------|------------------------------|
| CoT | Acc | 34.0 | 62.1 | 50.0 | 43.0 |
| | Error | 0.3 | 0.0 | 0.0 | 0.0 |
| | Miss | 0.0 | 0.0 | 0.3 | 0.0 |
| Domain | Acc | 32.1 | 66.0 | 51.0 | 43.5 |
| | Error | 0.3 | 0.0 | 0.0 | 0.0 |
| | Miss | 0.1 | 0.0 | 0.0 | 0.0 |
| Emotion | Acc | 35.0 | 64.3 | 50.8 | 44.0 |
| | Error | 0.1 | 0.0 | 0.0 | 0.0 |
| | Miss | 0.5 | 0.0 | 0.0 | 0.0 |
| None | Acc | 34.3 | 64.4 | 49.6 | 43.4 |
| | Error | 0.5 | 0.0 | 0.0 | 0.0 |
| | Miss | 0.4 | 0.0 | 0.0 | 0.0 |
| Rhetoric | Acc | 33.4 | 63.0 | 49.3 | 43.4 |
| | Error | 0.3 | 0.0 | 0.0 | 0.0 |
| | Miss | 0.3 | 0.0 | 0.0 | 0.0 |

Table 10: Accuracy, Error and Miss rate of different models under different settings.(2/4)

| Mode | Metric | GLM-4V-9b | LLaVA-1.6-72B | LLaVA-1.6-34B | idefics2-8b |
|---|---|---|---|---|---|
| CoT | Acc | 49.1 | 45.3 | 44.5 | 33.3 |
| | Error | 0.0 | 0.0 | 0.0 | 0.0 |
| | Miss | 0.0 | 0.0 | 0.0 | 0.0 |
| Domain | Acc | 49.9 | 47.3 | 46.4 | 37.5 |
| | Error | 0.0 | 0.0 | 0.0 | 0.0 |
| | Miss | 0.0 | 0.0 | 0.0 | 0.0 |
| Emotion | Acc | 51.1 | 48.6 | 47.1 | 38.6 |
| | Error | 0.0 | 0.0 | 0.0 | 0.0 |
| | Miss | 0.0 | 0.0 | 0.0 | 0.1 |
| None | Acc | 50.3 | 48.0 | 46.0 | 36.3 |
| | Error | 0.0 | 0.0 | 0.0 | 0.0 |
| | Miss | 0.0 | 0.0 | 0.0 | 0.0 |
| Rhetoric | Acc | 49.5 | 45.4 | 45.4 | 37.4 |
| | Error | 0.0 | 0.0 | 0.0 | 0.0 |
| | Miss | 0.0 | 0.0 | 0.0 | 0.0 |

Table 11: Accuracy, Error and Miss rate of different models under different settings.(3/4)

| Mode | Metric | Gemini-1.5 Pro | GLM-4V | GPT-4o | Claude-3-5-Sonnet | Qwen-VL-MAX |
|---|---|---|---|---|---|---|
| CoT | Acc | 54.1 | 49.9 | 54.9 | 51.6 | 54.8 |
| | Error | 0.3 | 3.4 | 0.0 | 1.8 | 1.1 |
| | Miss | 1.8 | 2.4 | 0.1 | 0.0 | 0.0 |
| Domain | Acc | 59.0 | 60.4 | 55.4 | 56.4 | 59.1 |
| | Error | 0.3 | 1.6 | 0.0 | 2.5 | 1.5 |
| | Miss | 1.4 | 0.0 | 0.0 | 0.0 | 0.1 |
| Emotion | Acc | 58.0 | 60.6 | 54.9 | 53.5 | 59.9 |
| | Error | 0.3 | 3.4 | 0.0 | 2.5 | 1.1 |
| | Miss | 1.8 | 0.0 | 0.1 | 0.0 | 0.0 |
| None | Acc | 60.1 | 60.9 | 54.1 | 54.1 | 56.9 |
| | Error | 0.3 | 0.0 | 0.0 | 3.3 | 1.9 |
| | Miss | 0.1 | 0.0 | 0.0 | 0.9 | 0.0 |
| Rhetoric | Acc | 55.6 | 58.8 | 51.9 | 54.9 | 54.8 |
| | Error | 0.3 | 2.1 | 0.0 | 1.9 | 0.9 |
| | Miss | 0.9 | 0.0 | 0.1 | 0.0 | 0.0 |

Table 12: Accuracy, Error and Miss rate of different models under different settings.(4/4)

## H    OTHER ERRORS

**Lack of Cultural Background Knowledge (16%):**

CII-Bench requires a model's deep understanding of Chinese traditional culture. Lacking knowledge of traditional symbols, historical figures, and classical allusions, GPT-4o struggles with interpreting deeper implications within images. Despite reasonable Chinese language handling, the model's cultural deficiency affects its reasoning and performance.

**Superficial Reasoning (12%):**

Understanding extended meanings within images is crucial. However, GPT-4o often only focus on surface-level elements, neglecting the deep implications and deeper cultural connotations behind them. This superficial reasoning hinders the model from fully grasping profound messages that the artist or designer intends to convey.

**Misunderstanding of Visual Information (11%):**

Accurate identification of visual information is vital. We find that GPT-4o sometimes misidentifies visual elements within images, particularly when dealing with abstract images. The abstract nature of such images often stems from the inclusion of exaggerated imaginative elements, sometimes even defying physical laws. Therefore, correctly identifying these abstract elements requires the model to have a deep understanding of the essence of objects, a capability that current models clearly do not yet possess.

# I CASE STUDY

The appendix is our sample analysis of GPT-4o, including an analysis of six error examples.

**List of Case Study Figures**

**Textual Information Neglect**

**Question:** 这张图片有什么隐喻？ <image 1>
**Option:**
(A) 禁止饲养宠物
(B) 禁止拍摄鸟类
(C) 禁止飞行
(D) 禁止发出大的声音
(E) 禁止饲养珍惜动物
(F) 禁止传播色情内容

<image 1>

**GPT-4o:** 让我们逐一分析每个选项：

(A) 禁止饲养宠物 - 该图片显示一个被禁止符号（红色圆圈和斜线）覆盖在一只鸟的图像上，并且无标识任何宠物特定的信息。
(B) 禁止拍摄鸟类 - 图片并没有显示任何与拍摄相关的符号或装置。
(C) 禁止飞行 - 虽然鸟可以飞，但图片没有特别强调飞行的元素。
(D) 禁止发出大的声音 - 图片没有显示与声音相关的符号。
(E) 禁止饲养珍惜动物 - 该选项与禁止饲养宠物类似，但没有特别标识珍惜动物的信息。
(F) 禁止传播色情内容 - 这完全与图像无关。

综合来看，这张图片最直接的解释是禁止养鸟。

答案：A

**Error Reason:** 图片中展示了一只鸟，一个禁止符号，以及非常显眼的四个大字"禁止色禽"。这里的色禽谐音了色情，也指图片中颜色花哨的鸟，因此图片想要传达的是禁止色情。而GPT-4o却因为忽略了图片中"禁止色禽"，导致其对图片的深意理解错误。

**Ground Truth: (F) 禁止传播色情内容**

Figure G1: A sample error case of *Information Neglect*.
Back to List of figures

**Visual Information Neglect**

**Question:** 这张图片有什么隐喻？ <image 1>
**Option:**
(A) 这张图片反映了家庭内部女性受到了男性的欺凌，呼吁社会关注家庭暴力和男女平等。
(B) 红色领带可以被视为外界对个人身份的期望或定义，而女性手中散落的文件则象征着她内在的真实自我。
(C) 这张图片批判职场中女性受到权力的不平等的待遇。
(D) 这张图片展示了女性与巨大红色领带的互动，象征着个人在面对社会传统习俗时的挣扎。红色领带代表传统的束缚，而女性散落的文件则表现出她试图突破这些束缚的努力。
(E) 这张图片批判了职场中男性表现出的虚伪，因为蛇往往象征虚伪。
(F) 在这张图片中，一个身穿黑白衣服的女性站在巨大的红色领带前，领带如同一条巨蛇，暗示着人类与未知力量的斗争。

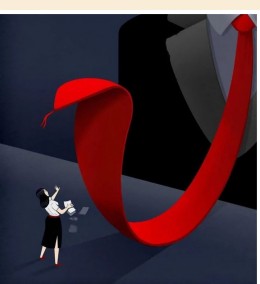

<image 1>

**GPT-4o:**为了逐一分析每个选项：
(A) 这张图片反映了家庭内部女性受到男性的欺凌，呼吁社会关注家庭暴力和男女平等。
- 这个选项略显偏颇，因为图片似乎并没有直接传达家庭暴力或家庭内部性别不平等的具体细节。
(B) 红色领带可以被视为外界对个人身份的期望或定义，而女性手中散落的文件则象征着她内在的真实自我。
- 确实可能与图片的一部分思想契合，即红色领带作为一种外界期望的象征，而散落的文件可能暗示女性内心的挣扎和真实自我。
(C) 这张图片批判职场中女性受到权力的不平等的待遇。
- 虽然这确实可能是一部分涵义，但图片中的元素（红色领带、散落的文件）更多地暗示社会期望和传统束缚的紧张关系。
(D) 这张图片展示了女性与巨大红色领带的互动，象征着个人在面对社会传统习俗时的挣扎。红色领带代表传统的束缚，而女性散落的文件则表现出她试图突破这些束缚的努力。
- 这个选项很切题，领带代表着传统和期望，而散落的文件体现了突破束缚的努力，符合图片的整体氛围和隐喻。
(E) 这张图片批判了职场中男性表现出的虚伪，因为蛇往往象征虚伪。
- 把领带比作蛇或虚伪有些勉强，图片中的领带更多的是象征某种外界期望或传统束缚。
(F) 在这张图片中，一个身穿黑白衣服的女性站在巨大的红色领带前，领带如同一条巨蛇，暗示着人类与未知力量的斗争。
- 提到"巨蛇"和"未知力量"有些离题，更多的是在描绘传统束缚的隐喻。
答案：D

**Error Reason:** 图片中红色领带呈现出了蛇的形象，暗示了职场中女性受到来自男性的权利威压，批判职场中女性受到权力的不平等的待遇。GPT-4o没有认出领带的形状，认为领带代表传统的束缚，这显然不合理，散落的文件也并非代表着努力。总之，GPT-4o由于忽略了视觉信息，导致了对图片的主题理解错误。

**Ground Truth: (C) 这张图片批判职场中女性受到权力的不平等的待遇。**

Figure G2: A sample error case of *Information Neglect*.
Back to List of figures

Figure G3: A sample error case of *Over-Inference*.
Back to List of figures

## Lack of Cultural Background Knowledge

**Question:** 这张图片有什么隐喻？<image 1>
**Option:**
(A) 老人可能代表着即将放手的母亲，年轻的女性则是即将成为家庭一部分的新成员。中间的男性正处在人生的重要转折点上，需要学会独立。
(B) 这张图片描绘了一个普通的购物场景。
(C) 这张图片展示了家庭外出旅游的场景。
(D) 老人手中的篮子象征着她对年轻一代的教导和支持；年轻女性的时尚装扮则代表了新思想的引入。中间的男性则体现了作为桥梁的作用，连接着过去与未来。
(E) 这张图片形象展示了对婆媳之间关系的调侃，表面和平而实际很分裂。
(F) 这张图片生动地展现了婆媳之间的和谐共处与相互支持的主题。画面中的三人手牵手，象征着家庭成员间的团结与和睦。

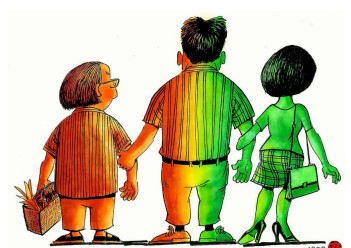

<image 1>

**GPT-4o:**让我们逐一思考每个选项。
(A) 老人可能代表着即将放手的母亲，年轻的女性则是即将成为家庭一部分的新成员。中间的男性正处在人生的重要转折点上，需要学会独立。
- 梳理选项中的描述，老人可代表一代人，年轻女性代表新一代，男性则是连接两代人的桥梁，中间的男性正处于转折点，这与图片中的人物关系和手牵手的方式似乎匹配。
(B) 这张图片描绘了一个普通的购物场景。
- 从图片来看，没有明显的购物元素。
(C) 这张图片展示了家庭外出旅游的场景。
- 图片中没有明示旅游的细节或装饰，没有明显的展现旅游场景。
(D) 老人手中的篮子象征着她对年轻一代的教导和支持；年轻女性的时尚装扮则代表了新思想的引入。中间的男性则体现了作为桥梁的作用，连接着过去与未来。
- 分享了明确的分析和隐喻，老人手中的篮子象征支持，年轻女性代表新思想，桥梁则表示连接。这与图像所传达的有较好的一致性。
(E) 这张图片形象展示了对婆媳之间关系的调侃，表面和平而实际很分裂。
- 图片没有表现出明显的分裂和矛盾。
(F) 这张图片生动地展现了婆媳之间的和谐共处与相互支持的主题。画面中的三人手牵手，象征着家庭成员间的团结与和睦。
- 表述与图像中的人物关系较为吻合，表达了团结与和睦。
根据以上分析，更合适的选项为 (D)，较为全面地解释了图片中的隐喻。
答案：D

**Error Reason:** 图片中展示了3个人物手拉手，居中的人是丈夫，右边是妻子，左边则是婆婆。婆婆呈现红色，妻子呈现绿色，而丈夫则是红绿相间。婆婆和妻子两种截然不同的颜色，暗示了她们之间的冲突，讽刺了当代社会中婆媳之间的冲突。丈夫被冲突所波及，左右为难。GPT-4o缺乏对中国文化的理解，因此无法选出正确的答案。

**Ground Truth: (E) 这张图片形象展示了对婆媳之间关系的调侃，表面和平而实际很分裂。**

Figure G4: A sample error case of *Lack of Cultural Background Knowledge*.

## Superficial Reasoning

**Question:** 这张图片有什么隐喻？ <image 1>
**Option:**
(A) 两者共同拉起横幅，意味着传统与现代技术在对抗浪费方面的合作。人群象征着团结的力量。
(B) 在这张图片中，直升机和起重机代表着不同形式的力量汇聚在一起，共同为反对浪费的目标努力。人群展示了大众对于这一事业的热情和支持。
(C) 图片反映了现代社会信息传播的困境：信息爆炸的同时，有效信息的获取和理解却变得越来越困难。
(D) 直升机代表着政府的支持，而起重机则象征着社会基层的努力。横幅上的"反铺张浪费万人大签名"表明人们希望通过大型活动来提高对节约资源意识的关注。
(E) 这张图片对讽刺了目前因为反对铺张浪费现象而产生的不合理行为。
(F) 图片赞扬了广大人民群众为了共同的目标而努力的合作精神。

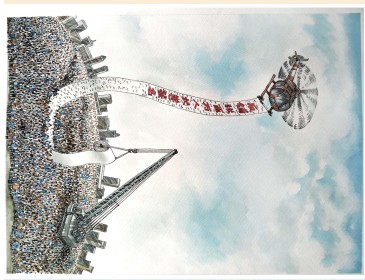 <image 1>

GPT-4o:从这张图片中，可以看到直升机和起重机共同举起了一个标有"反铺张浪费万人大签名"的横幅，同时下面有大量的人群集结。各个选项的分析如下：
(A) 两者共同拉起横幅，意味着传统与现代技术在对抗浪费方面的合作。人群象征着团结的力量。
- 图片中直升机和起重机确实共同举起了横幅，人群也显示了团结的力量。这种描述贴近图片内容。
(B) 在这张图片中，直升机和起重机代表着不同形式的力量汇聚在一起，共同为反对浪费的目标努力。人群展示了大众对于这一事业的热情和支持。
- 这种说法也与图片内容相符，但没有明确现代与传统技术分别指代什么形式的力量。
(C) 图片反映了现代社会信息传播的困境：信息爆炸的同时，有效信息的获取和理解却变得越来越困难。
- 此选项与图片所展示的内容不符，主要不涉及信息传播的困境。
(D) 直升机代表着政府的支持，而起重机则象征着社会基层的努力。横幅上的"反铺张浪费万人大签名"表明人们希望通过大型活动来提高对节约资源意识的关注。
- 图片中的直升机和起重机虽然显示了支持，但没有明显证据表明直升机代表政府、起重机代表基层。
(E) 这张图片讽刺了目前因为反对铺张浪费现象而产生的不合理行为。
- 从图片来看，没有表现出对不合理行为的讽刺倾向，主要是为了展示反对铺张浪费的行动。
(F) 图片赞扬了广大人民群众为了共同的目标而努力的合作精神。
- 人群众多，似乎确实展示了为了共同目标而合作的精神。
答案分析：综合来看，选项 (A) 和 (B) 都比较贴合图片内容，但 (A) 更为具体地说明了技术与团结的隐喻。因此选 (A)。
答案：A

**Error Reason:** 图片夸张的展示了反对浪费的万人签名活动。密密麻麻的人群、拉起签名纸的直升机和起重机，的确体现了人民的团结精神，但是这并非作者想要传达的思想。如此浮夸、耗费资源和人力的签名活动，正是一种浪费的体现，可笑的是活动本身却是反对浪费的。GPT-4o只意识到了最肤浅的图片含义，没有充分结合图片信息，挖掘出深意。

**Ground Truth: (E) 这张图片对讽刺了目前因为反对铺张浪费现象而产生的不合理行为。**

Figure G5: A sample error case of *Superficial Reasoning*.

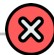 **Misunderstanding of Visual Information**

**Question:** 这张图片有什么隐喻? <image 1>
**Option:**
(A) 爱情让人们失去了理性判断。
(B) 爱让我们变得更加聪明和睿智。
(C) 爱是一种需要持续投入的投资。
(D) 爱情会导致人们的孤立和疏远。
(E) 过于充盈的爱, 会让我们失去独处的思考。
(F) 爱情会让人更加关注自己的内心世界。

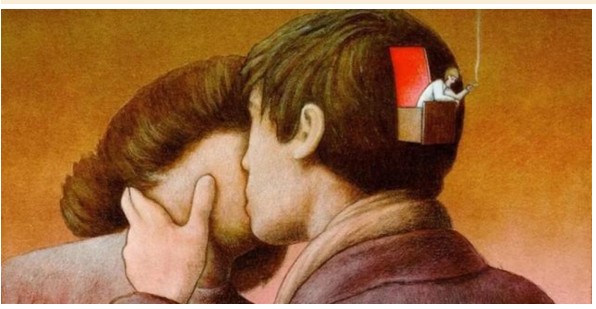

<image 1>

**GPT-4o:**这幅图片展示了一对年轻人亲吻的场景, 其中一个人的头脑中有一个操作杆的人, 这个细节表明隐喻的含义。逐个选项分析:

(A) 爱情让人们失去了理性判断。
- 操作杆的人可能暗示爱情在操控一个人的理智, 显示出理性被放在一个小角落来观察, 可以一定程度上支持这个观点。
(B) 爱让我们变得更加聪明和睿智。
- 从图片来看, 这种解释不太符合, 因为图中的操控体现了某种控制, 而不是增强智慧。
(C) 爱是一种需要持续投入的投资。
- 没有直接暗示投入或投资的行为, 所以这个选项不太符合。
(D) 爱情会导致人们的孤立和疏远。
- 图片中两个人紧密接触, 并没有表现出孤立和疏远的感觉。
(E) 过于充盈的爱, 会让我们失去独处的思考。
- 这也不符合图片中的暗示, 图中没有表现出爱太多导致失去独处思考的情境。
(F) 爱情会让人更加关注自己的内心世界。
- 操作杆的人是在脑中, 可能象征着对内心世界的观察和控制, 但更接近的是对于理性判断的控制, 而不是单纯的内心世界关注。

综上所述:

答案: (A)

**Error Reason**: 图片展示了一对男女接吻, 奇怪的是男子的后脑有一个小阳台, 阳台上有个小人正在抽烟。抽烟往往暗示着思考, 图片可能想要暗示充斥着的爱让我们没有独立思考的时间与空间。GPT-4o认为图中男子后脑有一个操纵杆, 这显然是错误的。正是由于错误的识别为操纵杆, 才导致它的分析全部错误。

**Ground Truth: (E) 过于充盈的爱, 会让我们失去独处的思考。**

Figure G6: A sample error case of *Misunderstanding of Visual Information*.

