# OpenReview forum: "Can MLLMs Understand the Deep Implication Behind Chinese Images?"
_ICLR.cc/2025/Conference — ICLR 2025 Conference Withdrawn Submission_

### Official Review · Reviewer_LmXG · 2024-10-26

**Soundness:** 3
**Presentation:** 4
**Contribution:** 2
**Rating:** 5
**Confidence:** 5

**Summary:**

In this paper, the authors construct a benchmark called CII-Bench, designed to measure the high-order perception and understanding abilities of MLLMs for Chinese images. CII-Bench consists of 698 images and 800 multiple-choice questions, spanning six different domains. To construct the dataset, the authors: i) collected 17,695 images from various websites, ii) filtered these images to retain only those suitable for the benchmark's goals, and iii) manually annotated the remaining images with detailed instructions. After constructing the dataset, the authors evaluated various models, both open-sourced and proprietary, to assess the capability of existing MLLMs in understanding the deep implications of Chinese images. Additionally, the authors employed different prompting strategies, such as Chain-of-Thought (COT) and few-shot learning, to fully explore the potential capabilities for this type of task.

**Strengths:**

1. This dataset is constructed using a rigorous pipeline that includes repeated image filtering and consistency checks, ensuring its high quality.
2. The number of models used for evaluation is extensive, encompassing both open-source and proprietary options, and we can observe a significant performance gap between different models.
3. Compared to most previous works, the prompting strategies used for evaluation are quite exhaustive, making the results highly informative and instructive.
4. This paper is well-written and easy to read.

**Weaknesses:**

1. The size of this dataset—698 images and 800 questions—is quite small, which may render the conclusions drawn from the evaluation results non-generalizable.
2. Since all the questions in this benchmark are multiple-choice, the output obtained from MLLM may be biased, as some models tend to favor specific choices. Therefore, the authors are encouraged to use the ``CircularEval`` in MMBench[^1] to ensure more robust results.
3. As shown in Table 1, text-only models, such as Qwen2-7B-Instruct, can also answer some questions correctly without referring to images. As a benchmark for evaluating multimodal capabilities, these questions are not appropriate and should be removed.
4. Lacking comparison with CCBench[^2], a benchmark designed to evaluate an MLLM's capability to understand images related to Chinese culture.
5. The benchmark primarily evaluates a model's ability to understand the implications behind Chinese images. However, I find that some images in the appendix, such as Figure G3 and Figure G4, are not particularly representative of Chinese imagery.
6. Since the dataset is divided into six categories, the authors are also expected to explain the rationale behind choosing these specific categories.

[^1] https://arxiv.org/abs/2307.06281
[^2] https://github.com/open-compass/MMBench

**Questions:**

see weakness

---

### Official Review · Reviewer_PXtt · 2024-10-31

**Soundness:** 3
**Presentation:** 3
**Contribution:** 2
**Rating:** 3
**Confidence:** 5

**Summary:**

The author introduces the Chinese Image Implication Understanding Benchmark (CII-Bench), which aims to evaluate the advanced perception and understanding capabilities of multimodal large language models (MLLMs) for Chinese images. The author designs a data curation process and evaluates the proposed benchmark on multiple MLLMs.

**Strengths:**

- CII-Bench is intriguing, and its construction process is clearly presented, offering value for the development of image implication understanding.
- Evaluations are conducted on multiple open- and closed-source MLLMs, providing detailed analyses of CII-Bench from various perspectives.

**Weaknesses:**

- The proposed CII-Bench includes a greater emphasis on understanding the cultural and emotional content behind images. In this context, did the authors design more complex prompts to better guide the model's output? For instance, did they use background information and Chain-of-Thought (CoT) prompting to help the MLLM predict answers from the background context?

- The English images presented in Fig.~1 are not convincing, as there are also complex and suggestive English images. The authors should compare with similar datasets. For example, in the II-Bench work, there is already a significant gap between the performance of existing MLLMs and human results, which is highly consistent with the conclusions of this paper. The authors should provide more rigorous reasons to explain why Chinese images present unique challenges compared to English images.

- Using the II-Bench approach, the authors replicated the entire process on Chinese data. I did not see updated data collection and management content. The authors should distinguish their work from II-Bench in terms of scientific writing and benchmark processes. CII-Bench appears to be a derivative of II-Bench. The authors are advised to clarify additional contributions to enhance the innovation of this paper.

- To ensure the reliability of high-quality evaluation in CII, how did the authors consider the dataset size of less than 1K? Since CII covers multiple aspects, leading to fewer data points in each aspect, how did they address potential biases? The authors should provide statistically significant quantitative results to demonstrate the validity of CII-Bench.

- II-Bench introduced new challenges, and I look forward to seeing the authors' technical innovations to improve the benchmark results. Have the authors conducted relevant technical explorations, and from which perspectives will they address these issues?

**Questions:**

Please refer to the Weaknesses section.

---

### Official Review · Reviewer_V3pQ · 2024-11-07

**Soundness:** 3
**Presentation:** 3
**Contribution:** 2
**Rating:** 3
**Confidence:** 5

**Summary:**

(1) The authors introduced the Chinese Image Expression Understanding Benchmark (CII-Bench) aimed at evaluating MLLMs' ability to perceive and understand Chinese images at a high level.
(2) The authors found that MLLMs perform worse on Chinese and traditional cultural images, which suggests a limitation in their ability to recognize high-level images.

**Strengths:**

(1) This paper is the first benchmark work to propose Chinese image representation understanding, which is of some help to the multimodal large language model for understanding Chinese images.

(2) The paper comprehensively compares the capability of existing multimodal large language models.

**Weaknesses:**

**Weakness 1** It is mentioned in the paper that “in order to ensure the authenticity of the Chinese context, the pictures in CII-Bench are all from the Chinese Internet and have been manually reviewed, and the corresponding answers are also manually produced.” So the pictures are all from the Internet, and most of them are not real pictures, which greatly limits the development of Chinese language, and it is suggested that Chinese pictures from some real scenarios should be added.

**Weakness 2**  For some metaphorical work such as FigureG1, these answers are too simple, and the complexity should be increased.

**Weakness 3** It is suggested to add some datasets that contain more traditional Chinese culture, such as frescoes and landscape paintings, and experts are needed to judge and calibrate the labels.

**Weakness 4** This paper and reference [1] can clearly be combined into a dataset, as the constructed prompts are the same, with at most only some differences in the images. Therefore, for a top conference like ICLR, the contribution is relatively small.

> [1] II-Bench: An Image Implication Understanding Benchmark for Multimodal Large Language Models

**Questions:**

See weakness.

---

### Official Review · Reviewer_mjrp · 2024-11-08

**Soundness:** 2
**Presentation:** 3
**Contribution:** 2
**Rating:** 3
**Confidence:** 4

**Summary:**

This paper introduces the Chinese Image Implication understanding Benchmark, CII-Bench, to evaluate the capacity of MLLMs for higher-order perception and understanding of Chinese visual content. Through the experiments, the authors find the shortcomings of MLLMs in understanding Chinese visual content. Overall, this work is interesting.

**Strengths:**

- This paper is well-written and easy to read.
- Authors evaluate the performance of many different MLLMs. Generally speaking, the experiments are extensive.

**Weaknesses:**

- The scale of this dataset is a little small. CII-Bench only contains 698 images and 800 questions, which may not be comprehensive enough to evaluate the performance of MLLMs.
- Some detailed information about the dataset should be provided. For example, the ratio of six different types of images.
- The motivation is not strong enough. I think this work is just an extension of II-Bench [1]. So, to demonstrate the necessity of this paper, the authors should discuss or conclude the inconsistencies of the results on II-Bench and CII-Bench.
Since this paper is quite similar to II-Bench [1], it is important to analyze the consistencies and inconsistencies of the experimental results. For example, in which scenario do the MLLMs exhibit similar performance for images in two different languages? Meanwhile, under which conditions do the Chinese images present stronger challenges than the English ones?


[1] Liu, Ziqiang, et al. "II-Bench: An Image Implication Understanding Benchmark for Multimodal Large Language Models." arXiv preprint arXiv:2406.05862 (2024).

**Questions:**

- In the introduction, the authors emphasize that Chinese traditional landscape paintings may be more complex than English images. Based on this observation, the authors collect this dataset. However, CII-Bench contains many Meme images. Whether the Meme images in Chinese or English have large differences?
- In the experiments, authors find there exists a gap between humans and MLLMs. So, can you give some suggestions for future research to enhance the MLLMs and promote the performance of MLLMs in this field?

---

### Official Review · Reviewer_eNXR · 2024-11-11

**Soundness:** 3
**Presentation:** 3
**Contribution:** 3
**Rating:** 6
**Confidence:** 2

**Summary:**

This paper introduces a new benchmark, CII-Bench, for evaluating MLLMs on understanding Chinese image implications, which is an important capacity for MLLMs in achieving AGI. Qwen2-VL-72B achieves the best results of 64.4% accuracy but is still far away from human performance (78.2% on average). It also provides some insightful findings, e.g., models perform significantly worse in Chinese traditional culture compared to other domains. I believe this benchmark is valuable to the research community.

**Strengths:**

1. Understanding the Chinese image implication is an interesting and high-level capacity for MLLMs.
2. High quality of the dataset.
3. Sufficient analysis of experimental results.

**Weaknesses:**

1. The dataset is small.
2. Multi-choice evaluation may not reveal the real capacity to understand the implications.

**Questions:**

Do all samples use the same prompts (Figure 10) in CoT evaluation? It is strange that CoT and using few-shot examples got worse results.

---

### Note · Authors · 2024-12-02

**Comment:**

We appreciate the reviewers' time and detailed feedback. After carefully considering the comments, we have decided to withdraw our paper. \
Thank you once again for taking the time to review.

**Withdrawal Confirmation:**

I have read and agree with the venue's withdrawal policy on behalf of myself and my co-authors.